# Upscaling Customer Access Network Using Spectrum Conversion–Slicing–Duplication Technique

**Mohammad Syuhaimi Ab-Rahman [1,*] , Juwairiyyah Abdul Rahman [1,2], Nurul Farhana Mohd Arifin [1], Iszan Hana Kaharudin [3] and I-Shyan Hwang [4]**

1    Department of Electrical and Electronic Engineering, Faculty of Engineering & Built Environment, Universiti Kebangsaan Malaysia, Bandar Baru Bangi 43600, Selangor, Malaysia; juwairiyyah@unisel.edu.my (J.A.R.); a170175@siswa.ukm.edu.my (N.F.M.A.)
2    Faculty of Engineering, Universiti Selangor (UNISEL), Batang Berjuntai 45600, Selangor, Malaysia
3    School of Liberal Studies, Universiti Kebangsaan Malaysia, Bangi 43600, Selangor, Malaysia; iszanhana@ukm.edu.my
4    Department of Computer Science and Engineering, Yuan-Ze University, Chung-Li Dist., Taoyuan City 32003, Taiwan; ishwang@saturn.yzu.edu.tw
*    Correspondence: syuhaimi@ukm.edu.my

**Abstract:** The purpose of this study is to increase the number of access users without having to install a new optical cable. With the proposed solution, the cost of installation work can be reduced and the number of users can be increased. Several parameters were observed to ensure that the modified network not only improved the scalability but also met the standard parameters. Among the parameters observed are the Q factor, bit error rate (BER), eye diagram and number of users. The study was continued using the spectrum conversion, slicing and duplication technique, where the signals would be duplicated in arrayed waveguide grating (AWG) and sliced using a demultiplexer WDM (WDM Demux). Simulations were performed using the latest-version Optisystem 18.0 software by setting the transmitter frequency value of 1491 nm, transmitter power of 0 dBm and loss of 0 dB. The result shows the total user access achieved is 196,608 users. Meanwhile, the common FTTH network is allowed 256 users only. The criterion is set based on the calculation of the Q factor, which is greater than 6, while the BER is less than $1 \times 10^{-9}$. The Q factor for 196,608 users is 6.43617 and the BER is $4.52 \times 10^{-11}$. The number of users is increased without compromising the quality of data offered to the customer. Our solution is the first reported to date.

**Keywords:** scalability; conversion; slicing; duplication; access network





## 1. Introduction

GIGABIT-class passive optical networks (PONs) are currently being deployed in many countries because they have significantly higher capacity than traditional copper-based access networks. These networks allow for the delivery of broadband services, such as voice-over internet protocol (VoIP) and video on demand (VoD). There are two main gigabit-class PONs, the ethernet PON (E-PON (IEEE802.3ah)) and the gigabit PON (G-PON (ITU-T G.984)), which are standardized by the Institute of Electrical and Electronics Engineers (IEEE) and the International Telecommunications Union's Telecommunication Standardization Sector (ITU-T), respectively. In both standards, service is provided using time division multiplexing (TDM) technology, in which a point-to-multipoint (P2MP) connection is established between one optical line terminal (OLT) and several optical network units (ONUs). In an E-PON network, service is provided by one OLT over a 10–20 km feeder fiber to 16 ONUs, and 1 Gbp is symmetrically specified for the downstream and upstream traffic. In a G-PON network, service is provided over a 20 km feeder fiber between 1 OLT and 64 ONUs using asymmetric data rates of 2.5 Gbps for the downstream and 1.25 Gbps for the upstream. Because gigabit-class PONs provides an improved bandwidth segment

to each user compared with the copper-based access networks, they were envisioned as the ultimate solution to the problem of increased bandwidth demand. However, their capacity has been exhausted because of the newly developed high-bandwidth-consuming services, such as high-definition television (HDTV) and 3D television (3D-TV), as shown in Figure 1. For example, 10 Mbps is required to support a single HDTV channel, in addition to the bandwidth required for data and voice [1]. Thus, advancing from the gigabit-class PONs to next-generation optical access networks (NG-OANs) was inevitable. In addition to the high-speed requirement, the NG-OANs should overcome the other limitations of existing PONs, such as the low capacity, limited reach, and restricted mobility. Different architectures have been proposed in the literature to create an NG-OAN scheme that is able to satisfy particular sets of these limitations [2–5]. Significant effort was invested to unify the design of the ONU for the proposed colorless ONU, based on applying different techniques, such as injection locking, wavelength seeding (remote modulation), spectrum slicing, and wavelength tuning, so that each ONU is assigned a different wavelength [6]. The scalability feature refers to expansion of an optical access network to accommodate many users without compromising the bandwidth allocated [7]. A long-reach optical network is also a part of the scalability, in which the optical signal is converged to support long-range end users [8].

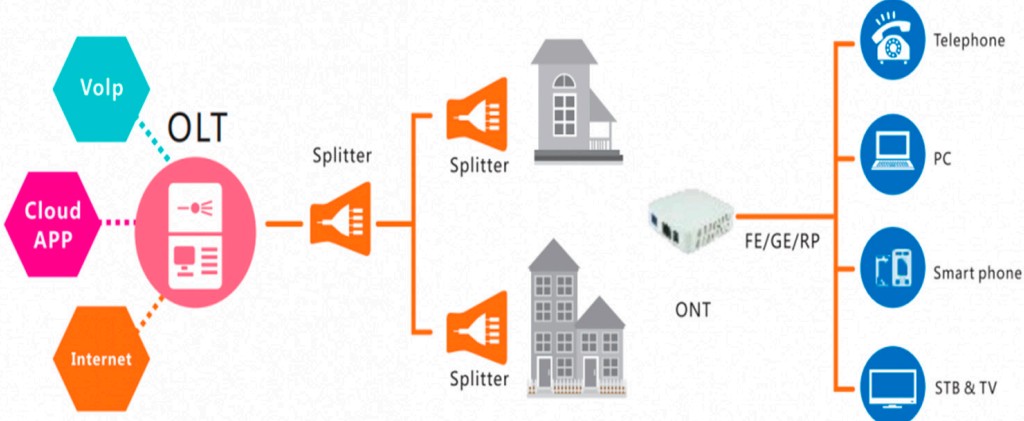

**Figure 1.** FTTH architectural illustration.

Most operators do not consider FTTH to be viable. They would rather extend their existing infrastructure. When rolling out a new network, the digging cost will play a decisive role, especially in rural areas. Important value is placed on the Capex cost per home passed (HP) [9]. Reducing the digging cost can significantly improve FTTH rollout. Combining digging works with planned road works, and using existing ducts with micropipes, are good ways to limit the digging cost. P2P technology is more expensive than P2MP if we able to reduce of the price of the digging cost, OLTs and ONUs.

The world has seen the incredible growth of the internet as a global means of communication. The FTTH network that we use today can reach only up to 256 users with the use of WDM Demux and an optical splitter with the ratio of 8 × 32 [10]. The first stage of research can expand that figure up to 6144 users with the addition of a new component in the network, AWG, with the latest ratio of 6 × 32 × 32. Then, the research continues to increase the scalability of the network with several additional components, which then expands the access of the users up to 196,608 users. This study was conducted how to increase the scalability of the fiber-to-the-home (FTTH) network until its maximum capacity without the need for new cable installation. The study was executed by adding some components to the network and measuring the bandwidth to produce not only the highest number of users but also a signal with an acceptable Q factor and bit error rate (BER). Through this study, we found there are numerous techniques that can increase the scalability of the network, maximize the usage of the internet cable, and also increase the efficiency of the existing cable network.

The technique that we use throughout this study is a spectrum conversion [5], slicing [11] and duplication [12] technique because it is the best technique for scalability as it is very efficient. Furthermore, the output separation ratio can be controlled and can be expanded to the maximum. As we are moving into the world of technologies, the demand for the internet is increasing significantly, and it did so especially during COVID-19. So, this study and research are really important in improving network upscaling and optimizing the efficiency of the existing cable network. Our proposed solution is able to support a huge number of users without deploying new cable installation.

## 1.1. AWG-Based Technique

Arrayed waveguide gratings (AWGs) do not require DWDM filters on each ONT, as shown in Figure 2. However, conceptually, using AWG outside of the plant comes with a host of issues. Since AWG is extremely sensitive to temperature variations, channel spacing is severely constrained. Furthermore, it is important to tune the DWDM source and keep an eye on the AWG path band's wavelength [13]. In order to create several WDM-based lightweight tree PONs on the same physical fiber, aggregated downstream traffic modulated at different wavelengths from the OLT is transmitted to the NxN AWG and the wavelengths are decompressed to its respective output ports. All ONUs in the light tree receive the wavelength from the AWG output port via broadcast using a $1 \times N$ optical divider. The ONU is then set to a particular fiber wavelength. By making use of the AWG FSR, an endless number of wavelengths can be transported from a certain input port to a particular output port. Service providers have the freedom to launch new services with little to no impact on their existing legacy systems thanks to the presence of many light trees in the same physical fiber link. Future scaling will be endless due to the physical PON's ability to support an infinite number of wavelengths. Furthermore, it utilizes passive NxN AWG and takes advantage of wavelength cycle routing made feasible by the device's free spectral range (FSR) at 2.5 Gbps. The unique characteristic of WDM-PON is the foundation for the suggested architecture, which includes the potential to simultaneously provide switch and broadcast services as well as additional benefits in terms of signal privacy, vulnerable location, and live capacity enhancement. It may also be possible to provide various types of services on physical networks. Optical transmission experiments reveal error-free transmission has been achieved using a 25 km passive optical link with $16 \times 16$ AWG and a $1 \times 32$ divider added, despite the additional AWG in the network having an additional 5 dB power penalty [14].

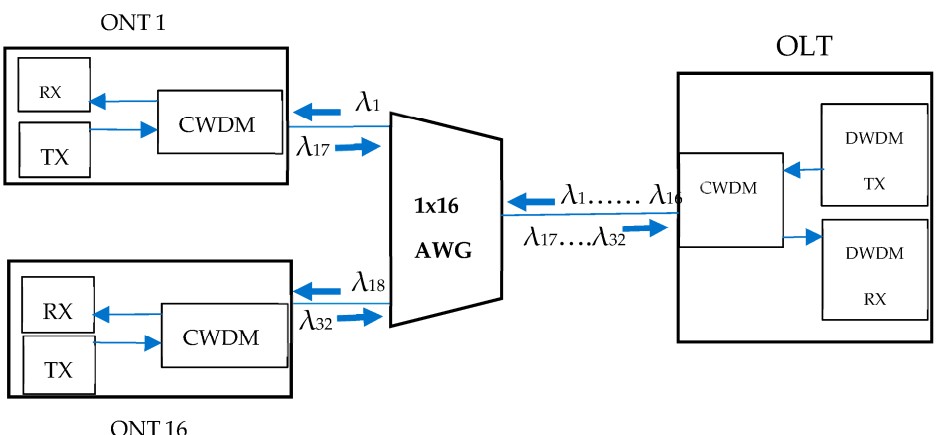

**Figure 2.** Technique using AWG.

## 1.2. EDFA-Based Technique

Without the use of a repeater, optical signals can be transmitted across great distances using erbium-doped fiber amplifiers (EDFAs). This enables the improvement of DWDM systems by incorporating extra sources at various wavelengths and DWDM multiplexing them into a single cable. Applications for WDM and fiber-to-the-home (FTTH) create

conflict and also present opportunities for the optical community. WDM operates at 1–40 GHz with channel separations as small as 0.8–0.4 nm in high-capacity situations. Advanced components are needed for this technology, such as a very stable wavelength source, precise optical filtering, cross-channel light management, and so on. Up until now, its network-access apps have been constrained. WDM is a very promising method for making use of the optical fiber's available bandwidth. However, due to the high cost of traditional WDM optical components, its usage for inclusion is restricted. The design of multiplexers and demultiplexers is more straightforward, and the tolerances are larger, in order to lower the price of optical transceivers and slow down the design criteria for WDM systems [15].

### 1.3. Hybrid System—OCDMA

The optical CDMA technique (OCDMA) is the solution for optical network access in the future, as depicted in Figure 3. The OCDMA technique allows multiple clients to simultaneously share the same frequency and time interval. There are two main types of CDMA optic technique, namely coherent and non-coherent. Coherent OCDMA codes use phase modulation, while non-coherent OCDMA codes are unipolar encoding systems that use amplitude modulation. A. Garadi has presented a new 2D encoding technique for the SAC-OCDMA system based on two orthogonal polarization conditions (vertical and horizontal) to increase the number of simultaneous users. The theoretical results show that the proposed 2D ZCC OCDMA system can double the number of users compared to the 1D ZCC OCDMA system [16]. The results show that for 10 users, the effect of beating noise and thermal noise is 10–22 watts/Hz and the error rate is $10^{-9}$. In addition, the researchers found that the designed system had an optimal transfer rate of 15 Gb/s for each user, reaching a length of 30 km of fiber.

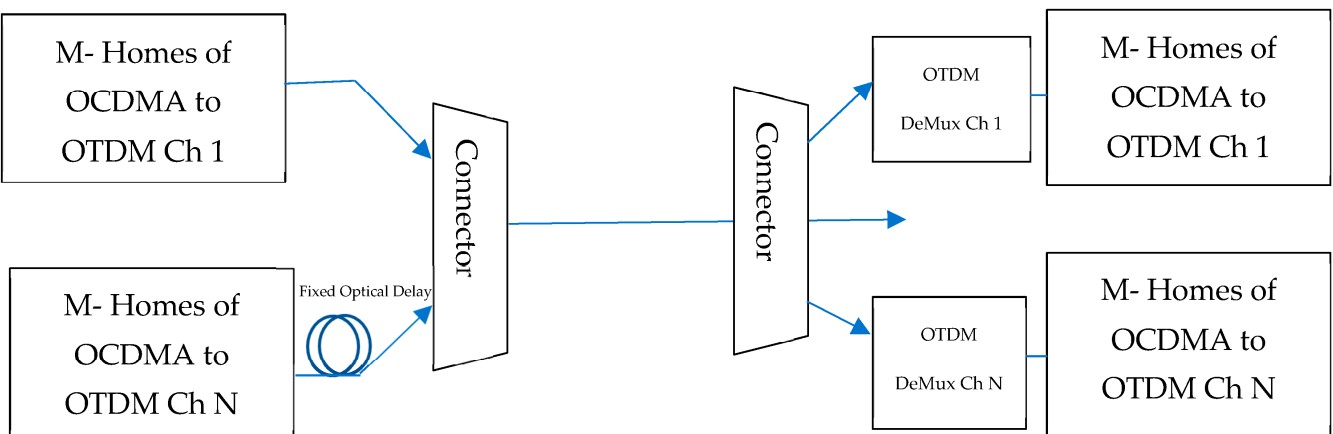

**Figure 3.** Hybrid system—OCDMA.

### 1.4. Hybrid System—10G-TDM-OCDMA-PON

The proposed 10G-TDM-OCDMA-PON hybrid system is based on the PON code interleaving TDM and 2D-MD (P/W) bits for the OCDMA stage. A downstream FTTH network of 16 houses and 4 high-speed TDM channels is simulated. In the OLT, the time division multiplexer for channel N has N number of users, where each time slot will be assigned to a particular ONU, the same multiplexer will be used with all TDM groups, and the only difference between them is the OCDMA encoder, where each TDM group has a different OCDMA code with different uses of wavelength and polarization. The TDM signal will modulate through a light wave encoded by the MZM, and then the modulated signal is grouped together by a power combine. On the receiver side, a power divider and a polarization reducer have been used to separate the polarized living signal. FBG is used as a diffusion compensator and OCDMA decoder to pass only the light signal assigned to a specific OTDMA group. A power divider is used to split the light signal for all the houses

and then an optical time division demultiplexer is placed inside each ONU to recover the information sent to this ONU [16].

## 2. Methodology

For this study, the FTTH network that has been modified was implemented using Optisystem 18.0 software simulation, to enable us to see the final signal that reaches the users and determine whether the signal is good or not. OptiSystem represents an optical communication system as a set of interconnected blocks. Each block is simulated independently using user-defined parameters for that block and signal information sent to it from other blocks. Through this software, we can gain experience of how to build the network, set the parameters, calculate the power and bandwidth and evaluate the eye diagram.

*Network Design*

In the addition of the existing components in the network, for network scaling, we add the EDFA, Mach–Zehnder modulator, AWG and power splitter (see Figure 4). The key in this study was to slice the broad-based signal into small spectrum before duplicate to many signals. The duplication will start at AWG, where it will then be broken down into multiple separator nodes. It is important to place components that can do the slicing and duplication early on in the network as it will increase the breakdown ratio and increase the final users. And as we carried out the simulation using Optisystem, we could later figure out the bit error rate (BER) and the Q factor.

The main objective for this study was to increase the scalability of the FTTH networks without the need to install a new optical cable. So, aimed to maximize the separation ratio with the help of some components, to increase the number of users that can be reached. The simulation results for the network implemented using the spectrum slicing and duplication technique showed a good signal at the end and also a maximum separation ratio for the AWG, power splitter, WDM Demux and power splitter, of $6 \times 32 \times 32 \times 32$, which reached approximately 196,608 users. Because of this addition of components (implementation of spectrum slicing and duplication technique), there is no need for installation of new optical cable; hence, we can lower the cost in increasing the scalability of networks. This creates an economical solution to solving high-density scenarios.

Instead of scalability, many research studies focus on the extension of the fiber length to support users located at distances far from the central offices [17–19] and effective bandwidth to support more application to the users [20,21].

The flowchart below (see Figure 5) shows how the simulation or the network works after the spectrum slicing and duplication technique is applied. This is to analyze the network scaling. In this study, the simulation for the upscaling was implemented using Optisystem software. To achieve the objective of the study, we changed the network by adding the AWG, power splitter, WDM Demux and power splitter. Using the new arrangement of the network, we collected the raw data of the signal going through the network, such as the maximum number of users, Q factor and bit error rate (BER).

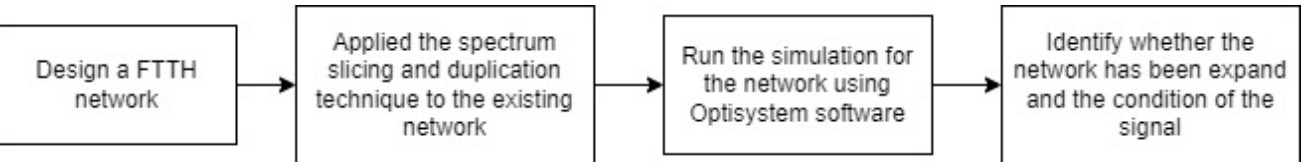

**Figure 4.** General design for network scaling using spectrum slicing and duplication techniques.

**Figure 5.** Flowchart of network scaling design and simulation.

## 3. Results and Discussion

### 3.1. Eye Diagram, Q Factor and Bit Error Rate (BER)

The eye diagram is an intuitive graphical representation of the signals of electrical and optical communication. From the appearance of the eyes, the quality of this signal can be observed. From the eye diagram, the bigger the area of the eye opening, the better the result. If the area of the eye opening is small, then it shows that the signal that reaches the users cannot be interpreted.

The signal-to-noise ratio (SNR) determines the quality of the received signal at a certain bit rate. The minimum Q factor is 6, where the signal must be equal or more than 6 to be said to have an SNR at the acceptable level. The Q factor for this network that reaches approximately 196,608 users is 6.43617, which is more than the minimum Q factor required. The other parameter that determines the performance of the optical network

is the bit error rate (BER). The BER is the number of bit errors per unit of time at the specific bit rate. The BER is the measurement of the number of bits that are errors during the transmission's completion, and the minimum requirement for the BER is $1 \times 10^{-9}$. Based on our simulation, our proposed network that accommodates 96,608 users has a BER = $4.52 \times 10^{-11}$, as shown in Figure 6.

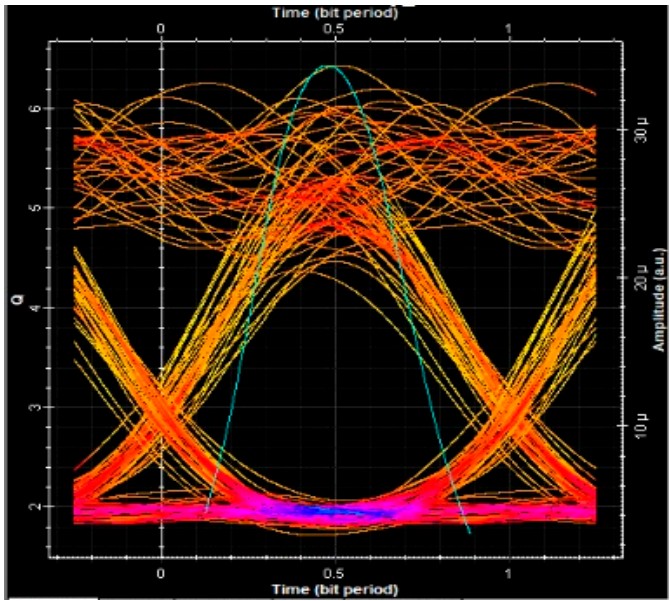

**Figure 6.** Eye diagram, Q factor and bit error rate.

### 3.2. Calculation for Number of Users

The main objective of this project is to increase the scalability by using the spectrum slicing and duplication technique. The maximum separation ratio that we can obtain is $6 \times 32 \times 32 \times 32$, which will reach approximately 196,608 users. The separation ratio represents the components or devices, which are the AWG, power splitter, WDM Demux and power splitter. This arrangement in the network has surpassed the maximum number of users for the existing network, which is 256 users.

### 3.3. Results for Power, Bandwidth and Group of Users

Tests were conducted to identify the appropriate bandwidth for the network. The bandwidths studied ranged from 100 GHz to 300 GHz. The following are the results of the Q factor and the bit error rate for each bandwidth. The AWG bandwidth is changed while the power and number of output ports (separation ratio) are the same, which are 0 dBm and $6 \times 32 \times 32 \times 32$ (196,608 users). From the data and graphs that have been plotted (Figure 7), it can be seen that the greater the bandwidth, the better the Q factor and the bit error rate (BER) value. This is because when the bandwidth is increased, the signal carried is high, making the final signal that reaches the user good, even if it has been sliced and duplicated. This theory can be described as the size of the pipe and the water that passes through it; when the size of the pipe is large, more water can pass through the pipe. Just like the analogy, the larger the bandwidth, the more signals or data that can pass through and can be transmitted at one time.

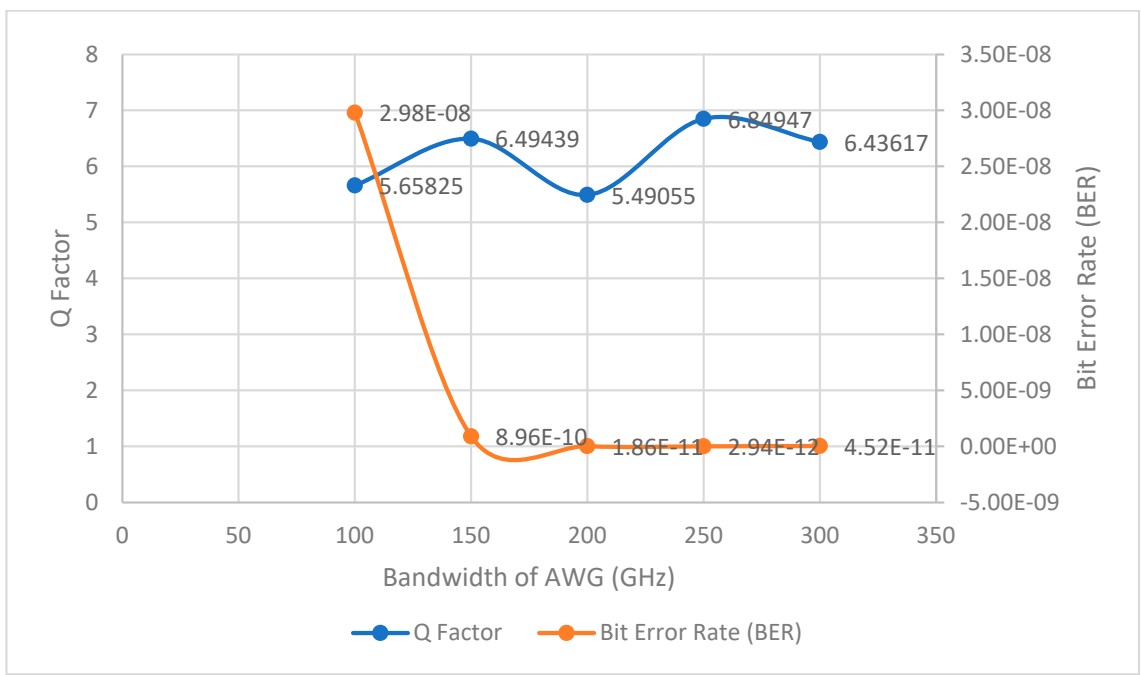

**Figure 7.** Bandwidth of AWG versus Q factor (number of users = 196,608).

After studying the user group as well as obtaining the appropriate bandwidth, a test was conducted to investigate the appropriate power for the network. The power studied ranged from 0 dBm to 4 dBm. The following are the results of the Q factor and the bit error rate for each power. The power is changed while the bandwidth and number of output ports are the same, which are 300 GHz and 6 × 32 × 32 × 32 (196,608 users). From Figure 8 that has been plotted, it can be seen that the more the power increases from 0 dBm to 4 dBm, the lower the value of the Q factor and the value of the bit error rate (BER). This is because when the optical power is too high, the 'gain coefficient' will decrease, which, in turn, will reduce the power of the signal. More precisely, when the optical power exceeds the saturation of the optical power (saturation optical power), the 'gain' will be saturated (saturated). This is related to the 'gain saturation' in EDFA, so the appropriate power for this network that uses EDFA is 0 dBm.

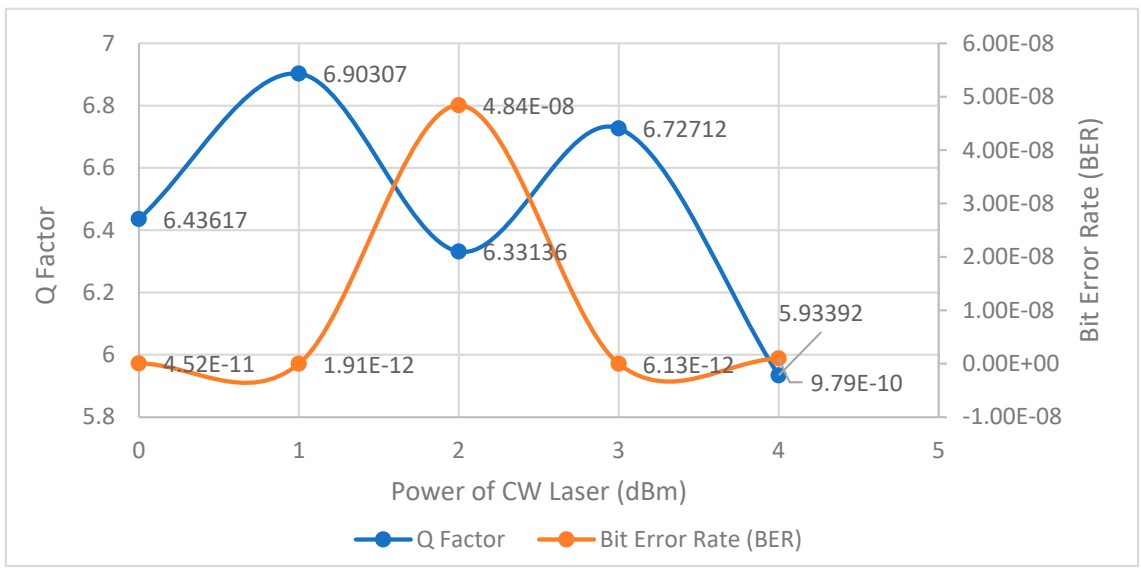

**Figure 8.** Power of CW laser versus Q factor (number of users = 196,608).

Once the appropriate power and bandwidth were determined, tests to confirm the best user group were conducted. The following are the data on the number of end users while ensuring that the end signal received is good. The bandwidth and power values are fixed at 300 GHz and 0 dBm.

Next, we tested the transmission rate of the network that had been modified while making sure the final signal received was acceptable. All other values that had been studied previously, such as bandwidth, power and splitting ratio values for the number of users (user groups), were fixed, i.e., at 300 GHz, 0 dBm and 196,608 users (AWG × power splitter × WDM Demux × power splitter: 6 × 32 × 32 × 32), as depicted in Figure 9.

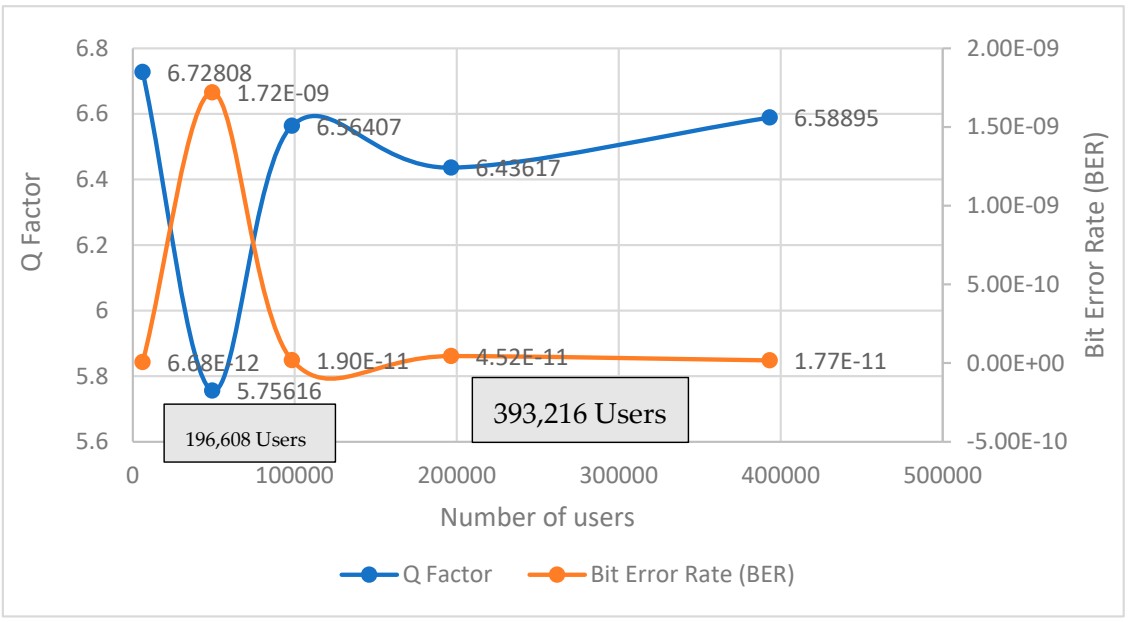

**Figure 9.** No. of users versus Q factor (bandwidth = 300 GHz).

### 3.4. Results for Maximum Allowable Users

The average Q factor was considered to determine the allowable number of users under a certain configuration. The Q factor is measured with respect to the change in the bandwidth for several targeted users.

Table 1 shows the Q factor versus the number of users at different bandwidths, from 100 GHz to 300 GHz. It seems that 196,608 users always fulfil the minimum allowable Q factor ($\geq$6). The average Q factor is 6.185766 (no. of users = 196,608), as compared to 5.892776 (no. of users = 393,216). Figure 10 depicts the average Q factors at different numbers of users. We can conclude the maximum number of users for this configuration (6 × 32 × 32 × 32) is 196,608. The Q factor for the user number (equal to 393,216) is less than 6; therefore, it cannot be considered for effective communication.

**Table 1.** Characteristics of Q factor in accordance with the number of users and bandwidth ($P_{in}$ = 0 dBm).

| No. of Users | 6144 Users | | | | 49,152 Users | | | | 98,304 Users | | | | 196,608 Users | | | | 393,216 Users | | | |
|---|---|---|---|---|---|---|---|---|---|---|---|---|---|---|---|---|---|---|---|---|
| | a | b | c | d | a | b | c | d | a | b | c | d | a | b | c | d | a | b | c | d |
| **Bandwidth (GHz)** | 6 | x | 32 | 32 | 6 | 8 | 32 | 32 | 6 | 16 | 32 | 32 | 6 | 32 | 32 | 32 | 6 | 64 | 32 | 32 |
| 100 | | 5.84612 | | | | 5.37531 | | | | 6.41122 | | | | 5.65825 | | | | 5.17017 | | |
| 150 | | 6.67228 | | | | 5.97101 | | | | 5.83785 | | | | 6.49439 | | | | 6.09388 | | |
| 200 | | 5.59458 | | | | 6.42985 | | | | 6.51374 | | | | 5.49055 | | | | 5.79648 | | |
| 250 | | 6.8909 | | | | 6.3323 | | | | 6.58316 | | | | 6.84947 | | | | 5.8144 | | |
| 300 | | 6.72808 | | | | 5.75616 | | | | 6.56407 | | | | 6.43617 | | | | 6.58895 | | |
| Q factor (Average) | | 6.346392 | | | | 5.972926 | | | | 6.382008 | | | | 6.185766 | | | | 5.892776 | | |

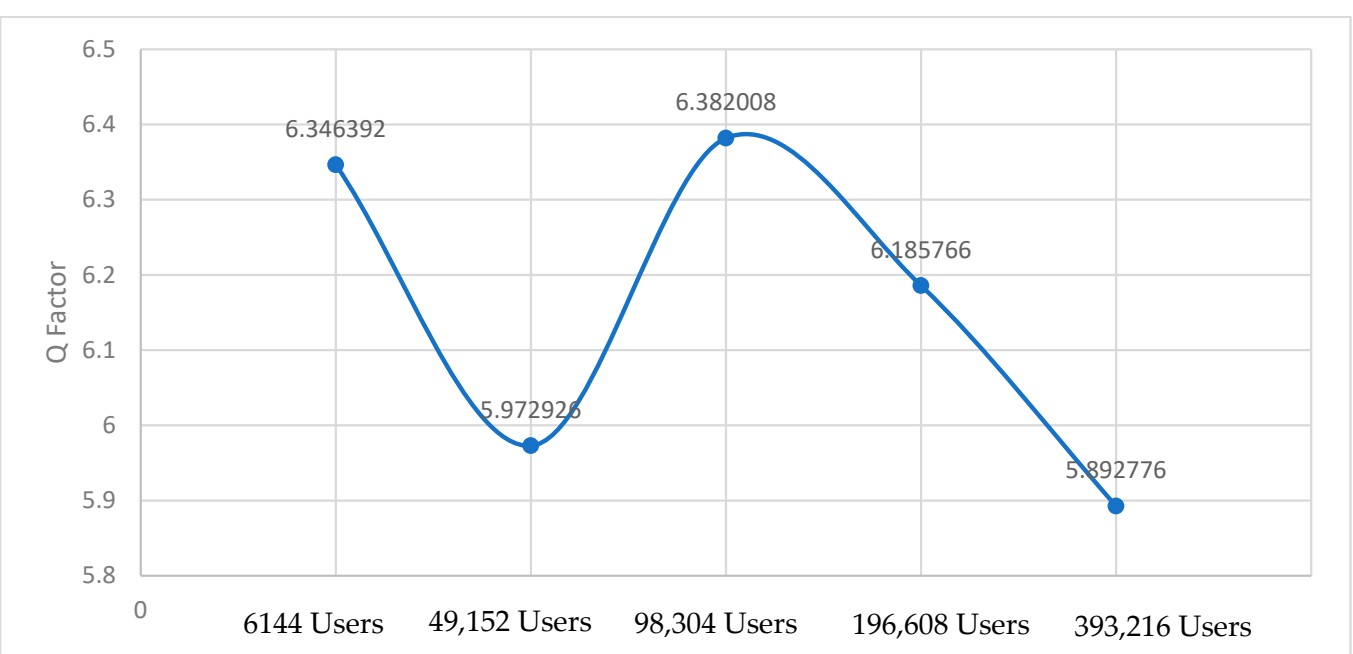

**Figure 10.** Max. Q factor (average) versus number of users.

## 4. Conclusions

This research explains how to increase the scalability of the existing network without installing new cable while ensuring the signals that reaches the end users are good and meet all the requirements that have been set (for Q factor and BER). There are a few ways of network scaling, such as XPM and XGM, or a hybrid technique like the OCDMA hybrid system. But, in this study, we choose the spectrum slicing and duplication technique as it is expandable. The simulation was conducted using Optisystem 18.0, where there are a few parameters that are observed, such as the eye diagram, Q factor and BER. Using this technique, the separation ratio can be controlled through the number of output ports of the components added to the network. The analysis was focused on this spectrum slicing and duplication technique and the results obtained. Through our study using this technique and network design, the total user access achieved was 196,608 users, as calculated through the output port of AWG $\times$ splitter $\times$ WDM Demux $\times$ splitter ($6 \times 32 \times 32 \times 32$). The Q factor for 196,608 users is 6.43617 and the BER is $4.52 \times 10^{-11}$, and it meets the standard for an optical communication system. This is the first reported simulation of the design to date.

**Author Contributions:** Conceptualization, N.F.M.A.; investigation, J.A.R.; supervision, M.S.A.-R.; review, I.H.K.; and editing, I.-S.H. All authors have read and agreed to the published version of the manuscript.

**Funding:** This research was funded by the Ministry of Science, Technology and Innovation (MOSTI), Malaysia, through the National Science Fund (NSF; 01-01-01-SF0493).

**Institutional Review Board Statement:** Not applicable.

**Informed Consent Statement:** Not applicable.

**Data Availability Statement:** Not applicable.

**Acknowledgments:** This research was conducted in the Broadband, Network & Security Laboratory, Universiti Ke-bangsaan Malaysia (UKM). The presented work was conducted in the Computer and Network Security Laboratory, Universiti Kebangsaan Malaysia (UKM), funded by the Ministry of Science, Technology and Innovation (MOSTI), Malaysia, through the National Science Fund (NSF; 01-01-01-SF0493).

**Conflicts of Interest:** The authors declare no conflict of interest.

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
