# Peer review of "Upscaling Customer Access Network Using Spectrum Conversion–Slicing–Duplication Technique"

_photonics, doi:10.3390/photonics10111271_

Round 1

Reviewer 1 Report

The author could add more recent references.

The authors could rewrite a few sentences to improve the quality of the text.

Author Response

Several articles have been added to the references list

Reviewer 2 Report

Reviewer report for Photonics- 2426062

“Upscaling Customer Access Network Using Spectrum Conversion-Slicing-Duplication Technique”

In this paper, an effort is made to present by means of simulation a technique to increase the number of access users without having to install a new optical cable, with acceptable Q-factor and bit error rate (BER) values. Although the treated subject is very interesting to the target scientific community, several critical issues which compromise the suitability of the paper for publication in Photonics in its current form. First of all, the information regarding the state-of-the-art is much confusing and misleading, as the authors use well-known techniques for the paper’s purpose. Second, the undertaken investigation is purely theoretical, without comparison to experimental evidence or careful consideration of implementation limits, which raises concerns about the practical feasibility of the scheme. Finally, in Figure 6 the Q-factor and BER values are not illustrated as reported by the author. Moreover, in Figures 8 and 9 the authors do not give some information about the fluctuations which are observed in some points of diagrams. For example in Figure 9, it is not so clear why we have a high Q-factor value for 50000 users (second point of the diagram).

No comments.

Author Response

1.     We used the simulation approach in getting the result. Due to the limitation of devices and components The simulation approach has been used to prove the feasibility of the project.

2.     500,000 is the estimated number of users, however, the BER and Q factor seem unacceptable (Q<6)

3.     The fluctuation is the tolerance in simulation in small-scale observation (floating point). The line can be considered straight and constant.

Reviewer 3 Report

This paper proposed a solution s to increase the number of access users without having 16 to install a new optical cable. The study was continued using the spectrum conversion, slicing and duplication technique where the signals would be duplicated in Arrayed Wave- 22 guide Grating (AWG) and sliced using Demultiplexer (DEMUX) WDM. But the results need more details.

Extensive editing of English language required.

Author Response

The English has been extensively checked and corrected

Round 2

Reviewer 2 Report

No comments. Accept in present form. Thank you.

No comments. Accept in present form. Thank you.